# Stem cell proliferation is induced by apoptotic bodies from dying cells during epithelial tissue maintenance

Courtney K. Brock[1], Stephen T. Wallin[1], Oscar E. Ruiz[1], Krystin M. Samms[1], Amrita Mandal[1], Elizabeth A. Sumner[1] & George T. Eisenhoffer [1,2]

Epithelial tissues require the removal and replacement of damaged cells to sustain a functional barrier. Dying cells provide instructive cues that can influence surrounding cells to proliferate, but how these signals are transmitted to their healthy neighbors to control cellular behaviors during tissue homeostasis remains poorly understood. Here we show that dying stem cells facilitate communication with adjacent stem cells by caspase-dependent production of Wnt8a-containing apoptotic bodies to drive cellular turnover in living epithelia. Basal stem cells engulf apoptotic bodies, activate Wnt signaling, and are stimulated to divide to maintain tissue-wide cell numbers. Inhibition of either cell death or Wnt signaling eliminated the apoptosis-induced cell division, while overexpression of Wnt8a signaling combined with induced cell death led to an expansion of the stem cell population. We conclude that ingestion of apoptotic bodies represents a regulatory mechanism linking death and division to maintain overall stem cell numbers and epithelial tissue homeostasis.

[1] Department of Genetics, The University of Texas MD Anderson Cancer Center, Houston, TX 77030, USA. [2] Genetics and Epigenetics Graduate Program, The University of Texas Graduate School of Biomedical Sciences at Houston, The University of Texas MD Anderson Cancer Center, Houston, TX 77030, USA. Correspondence and requests for materials should be addressed to G.T.E. (email: gteisenhoffer@mdanderson.org)

Epithelia serve as barriers that separate and protect our organs[1], regulate the transit of molecules[2,3], secrete cytokines[4] and perform a wide variety of specialized functions. As the first line of defense, the cells within epithelial tissues are constantly exposed to environmental insults that cause damage. Therefore, individual epithelial cells are continually being removed by apoptosis and replaced by proliferation of neighboring cells to retain the function and fitness of the tissue. Failure to efficiently coordinate the birth and death of cells can lead to dysregulation of cell numbers and compromised barrier function or, conversely, tissue hyperplasia and carcinoma formation. Yet, how cell death influences cell replenishment to fuel turnover during tissue homeostasis or after damage is not well understood.

Damaged cells targeted for elimination can influence the behavior of surrounding cells and have a dramatic impact on the form and function of epithelial tissues. Apoptotic cells in wing disc of *Drosophila*[5–7], Hydra after amputation[8], and amputated adult zebrafish fins[9] produce mitogenic signals that promote compensatory proliferation and regeneration. Activation of caspases during apoptosis is required for the production of the mitogenic cues in a wide variety of injury paradigms[10–12]. Among the cues produced in these different animal models are the members of the Wnt signaling pathway, known to play a key role in stem cell self-renewal during tissue repair and carcinogenesis[13–15]. However, further understanding is required to determine how damage induces the production and delivery of specific Wnts to mobilize stem cells for regeneration.

Apoptotic cells are rarely detected in vivo under physiological conditions, even in epithelial tissues with a high rate of turnover, owing to rapid clearance from the tissue[16]. Deconstruction of apoptotic cells incorporates membrane, cytoplasmic and nuclear constituents into small extracellular vesicles termed apoptotic bodies[17]. These extracellular apoptotic bodies range from 1 to 5 μm in size[18,19] and become enriched with proteins implicated in signal transduction, cell growth and maintenance[20]. However, the dynamic nature of the cell death and elimination processes has made studying the function of apoptotic bodies in living tissues difficult[16,21]. The limited ability to perturb and visualize these events in their native context has impaired our understanding of how cell death can stimulate proliferation to replace lost cells due to normal tissue maintenance or after serious injury.

The peripheral location and optical clarity of zebrafish epidermis make it very useful to monitor dynamic processes such as division, migration, extrusion and death of epithelial cells in vivo and in real time. The developing epithelium is bi-layered during embryonic and larval stages[22,23], consisting of cells in the basal layer that express the conserved epithelial stem cell marker p63[24–26] and periderm cells that comprise the apical surface[23,27]. The p63-positive cells in the basal layer serve as resident stem cells that self-renew and give rise to differentiated cytokeratin 8/18-positive periderm cells[28,29] and other specialized epithelial cell types such as ionocytes[30–32] in the outer surface layer. Further, dying outer keratinocytes in the zebrafish breeding tubercles stimulate compensatory proliferation of basal p63-positive cells to ensure homeostasis of this epidermal appendage[33]. Molecular tools[34–36] and imaging methods have already been established to interrogate specific cell types in the epidermis of the developing zebrafish, and gene function can be probed using both forward and reverse genetics[37–39]. Thus, the vast array of cell and molecular tools in zebrafish allow for investigation of the mechanisms guiding cell turnover in epithelial tissues with unprecedented resolution and control.

Here we induce damage in a subset of basal stem cells in the zebrafish epidermis and observe the dynamics of apoptotic cells as they are removed and replaced over time using time-lapse imaging. Using this approach, we uncovered that dying epithelial stem cells generate Wnt8a-containing apoptotic bodies in a caspase-dependent manner that stimulates proliferation of adjacent stem cells. Neighboring p63-positive stem cells rapidly engulf the apoptotic bodies, activate Wnt signaling and are subsequently stimulated to divide to maintain homeostatic cell numbers. Inhibition of either cell death or Wnt signaling abrogated the apoptosis-induced division, while overexpression of Wnt8a in combination with apoptosis led to a significant increase in overall cell numbers. Together, our results reveal a mechanism by which apoptotic cells transfer signaling molecules and induce localized proliferation to sustain cell numbers in a living epithelial tissue.

## Results

**Apoptosis induces epithelial stem cell proliferation.** To examine removal and replacement of stem cells within a living epithelium, we induced damage in a subset of basal stem cells in the zebrafish epidermis and observed the dynamics of apoptotic cells as they are removed and replaced over time using time-lapse imaging. We expressed the bacterial enzyme nitroreductase (*nfsB*, referred to as NTR) tagged with mCherry fluorescent protein[40,41], in approximately 75% of the p63-positive cells in the zebrafish fin epidermis using the *zc1036a* GAL4 enhancer trap line (Fig. 1a–c)[35]. Addition of the prodrug metronidazole (MTZ) to 4 days post-fertilization (4 dpf) larvae caused DNA damage (Supplementary Fig. 1a) and a rapid, dose-dependent increase in the number of activated caspase-3-positive cells expressing nitroreductase (Fig. 1d, e and Supplementary Fig. 1b, c). Apoptotic basal stem cells did not extrude via the apical layer in a manner similar to surface cells[34,42] or melanocytes[43], but became trapped between the basal and periderm layers and created noticeable bulges in the surface epithelium (Supplementary Fig. 1e). mCherry/activated caspase-3-positive cells were largely absent by 20 h after prodrug removal (Fig. 1f), indicating apoptotic cells are rapidly cleared from the tissue. These results demonstrate the ability to specifically induce apoptosis in a subset of p63-positive stem cells and establish a platform to observe cellular dynamics of the remaining cells that sustain epithelial tissue homeostasis.

We next sought to define how the increased apoptosis impacted the remaining stem cells in the basal layer. The density of the p63-positive stem cells significantly decreased after the wave of increased apoptosis, but homeostatic numbers were quickly restored over 24 h (Supplementary Fig. 2a-d). Accordingly, we observed a significant increase in the number of p63-positive cells undergoing DNA synthesis (Fig. 1g–i) or mitosis (Supplementary Fig. 2e-g) at 20 h after MTZ removal. Moreover, 97.87% of the BrdU+ cells analyzed after induced apoptosis were p63 positive. Time-lapse imaging also revealed actively dividing p63-positive basal cells after the wave of apoptosis (Supplementary Fig. 2h, i) and, taken together, suggest the dying basal stem cells induce a robust proliferative response (Fig. 1j).

To determine if apoptosis and caspase-3 activation were required for the observed proliferation, we examined the number of dividing cells after inhibition of cell death prior to and during induced apoptosis. Treatment of embryos with the Apoptosis Inhibitor II, NS3694, significantly reduced the number of active caspase-3-positive cells (Supplementary Fig. 1d) and resulted in a 41% decrease in stem cell proliferation (Fig. 1k). Importantly, caspase activity in regulating non-apoptotic processes was not affected using this approach. We next inhibited caspase-3 with the peptide inhibitor zDEVD-fmk and observed a similar decrease in active caspase-3-positive cells (Supplementary Fig. 1d) and stem cell proliferation (Fig. 1k). Cumulatively, these results suggest that caspase activation during apoptosis is required to induce stem cell divisions that restore cell numbers in the tissue.

**Dying stem cells produce Wnt-containing apoptotic bodies.**
Activation of caspases during apoptosis is required for the production of the mitogenic cues in a wide variety of injury paradigms[10–12]. Among the cues produced in these different animal models are the members of the Wnt signaling pathway, known to

play a key role in stem cell self-renewal during tissue repair and carcinogenesis[13–15]. Yet, the specific Wnt genes induced after epithelial stem cell damage remain largely uncharacterized. We examined components of both the canonical and non-canonical Wnt signaling pathways, as well as those previously shown to be

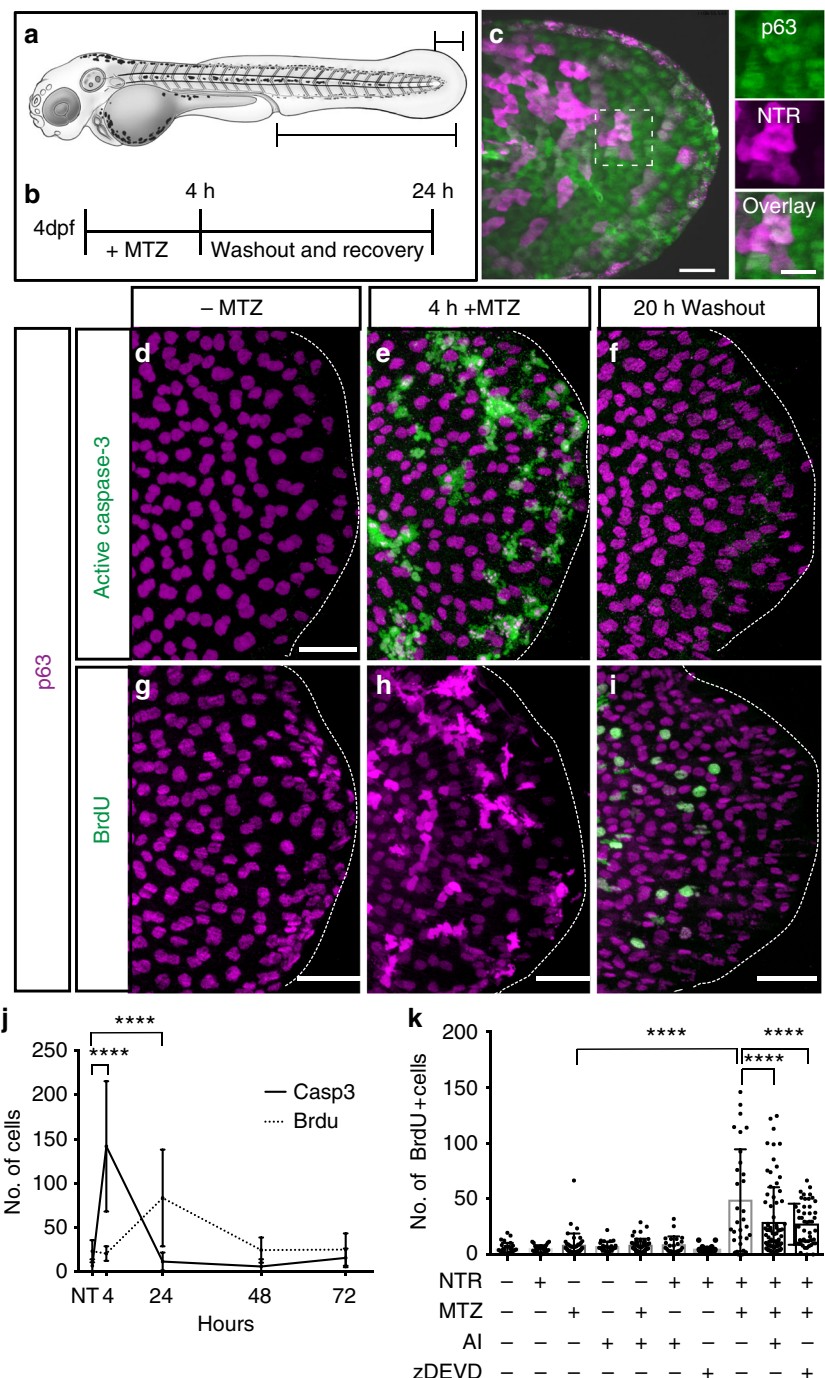

**Fig. 1** Caspase-dependent proliferation after stem cell ablation. **a** Schematic of a 4-day post-fertilization (dpf) zebrafish larvae. Large region denotes area of the animal where cell death and proliferation were quantified before and after cell ablation. Small region marks the area used for fixed and live imaging. **b** Timeline for the addition and removal of metronidazole (MTZ). **c** The *zc1036a* GAL4 enhancer trap line drives expression of fluorescently tagged nitroreductase (NTR) in a subset of p63-positive basal stem cells (scale = 100 μm, 50 μm inset). Maximum intensity projections of confocal images for activated caspase-3 (**d–f**) and bromodeoxyuridine (BrdU) (**g–i**) at different points after inducing damage (scale = 50 μm). **j** Quantification of active caspase-3- and BrdU-positive cells reveals a temporal relationship for the proliferative response. Mean number of positive cells from at least n = 11 individual larvae across three independent experiments per time point are plotted. NT=No treatment. **k** Mean number of BrdU-positive cells in individual larvae after induced apoptosis (n = 31) and in combination with treatment of the apoptosis inhibitor, NS3694 (AI) (n = 77) or caspase-3 peptide inhibitor zDEVD-fmk (zDEVD) (n = 49). Data are from three independent experiments and error bars represent sd; ****p < 0.0001. One-way analysis of variance (ANOVA) with Dunnett's mutiple comparisons test (**j**, **k**)

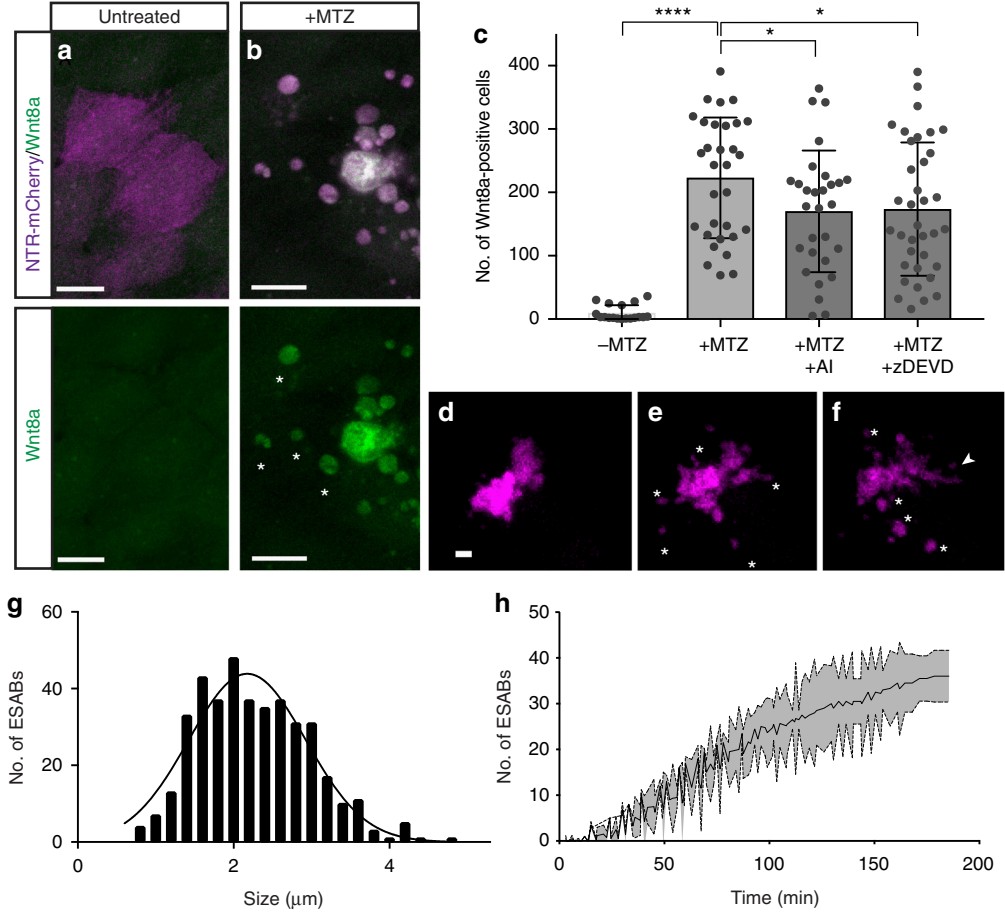

**Fig. 2** Caspase-dependent generation of Wnt8a in apoptotic cells and bodies. **a**, **b** Maximum intensity projections of confocal images of Wnt8a in healthy and apoptotic stem cells (scale = 10 μm). **c** Mean number of Wnt8a-positive cells in individual larvae after induced apoptosis (n = 31), and in combination with treatment of AI (n = 29) or zDEVD (n = 36). Data are from three independent experiments and error bars represent sd. **d–f** Still images from time-lapse microscopy of an individual apoptotic cell over time (scale = 5 μm), asterisks mark apoptotic bodies and arrowheads mark filopodia extensions (see Supplementary Movie 1). **g** Size distribution of epithelial stem cell-derived apoptotic bodies. **h** Quantification of the production of apoptotic bodies from individual cells. Data from at least three independent experiments are represented as mean ± sd. ****p < 0.0001, *p < 0.03, One-way analysis of variance (ANOVA) with Holm–Sidak multiple comparisons test (**c**)

involved in epithelial stem cells[44] or influenced by damage[45,46], after induced apoptosis to determine the particular signals that might promote the proliferation in our system. Wnt8a, as well as Wnt4b and Wnt11, showed a significant increase in mRNA levels after induction of apoptosis (Supplementary Fig. 3a). Interestingly, we observed an increase in Wnt8a specifically in the apoptotic cells (Fig. 2a–c and Supplementary Fig. 3b, c) with similar kinetics as activation of caspase-3 (Supplementary Fig. 3d, e), which was significantly reduced by treatment with NS3694 or zDEVD-fmk (Fig. 2c). These data indicate that Wnt8a is produced in the apoptotic cells in a caspase-dependent manner.

In addition to the apoptotic cell corpse, we also observed Wnt8a in small vesicular structures surrounding the dying cells (Fig. 2b, d–f). Apoptotic bodies are 1 to 5 μm extracellular vesicles that result from cytoskeletal reorganization and contraction that incorporates membrane, cytoplasmic and nuclear constituents during blebbing[17,19]. Time-lapse confocal microscopy of dying epithelial stem cells uncovered the generation of apoptotic bodies with an average diameter of 2.27 μm ± 0.71 (Fig. 2f, g and Supplementary Movie 1), with individual dying stem cells producing 12.87 ± 3.23 apoptotic bodies per hour (Fig. 2h). The apoptotic bodies derived from epithelial stem cells were similar in size and morphology to those observed from apoptotic rat thymocytes[47], T lymphocytes[48] and human endothelial cells[49].

Therefore, to distinguish from apoptotic bodies generated by other cell or tissue types, we refer to these as ESABs (*Epithelial Stem Cell-derived Apoptotic Bodies*). Filopodial projections, averaging 8.48 μm ± 2.19 in length, were also observed emanating from the apoptotic cells with ESABs located at the ends (Fig. 2e, f and Supplementary Fig. 4a–c). Retraction of the cellular extensions containing the F-actin marker Lifeact[50] appeared to promote detachment of the ESABs, often inside neighboring basal stem cells (Supplementary Fig. 4c). Membrane protrusions in apoptotic cells, known as apoptopodia, have been proposed to facilitate the separation of blebs into individual apoptotic bodies in cultured cells[20]. These data suggest that Wnt8a production in apoptotic cells is caspase dependent and that ESABs may provide a mechanism to transfer material and communicate with surrounding cells.

**ESAB engulfment stimulates Wnt signaling and proliferation.** To define the role of ESABs in the observed apoptosis-induced proliferation, we first sought to investigate the mechanisms used to clear the Wnt-containing apoptotic corpse and bodies from the tissue. Using time-lapse imaging, we observe p63:EGFP-positive stem cells engulfing multiple distinct apoptotic bodies (Fig. 3a–c and Supplementary Movie 2). Ultrastructural analyses confirmed the presence of 1 to 5 μm vesicular structures containing

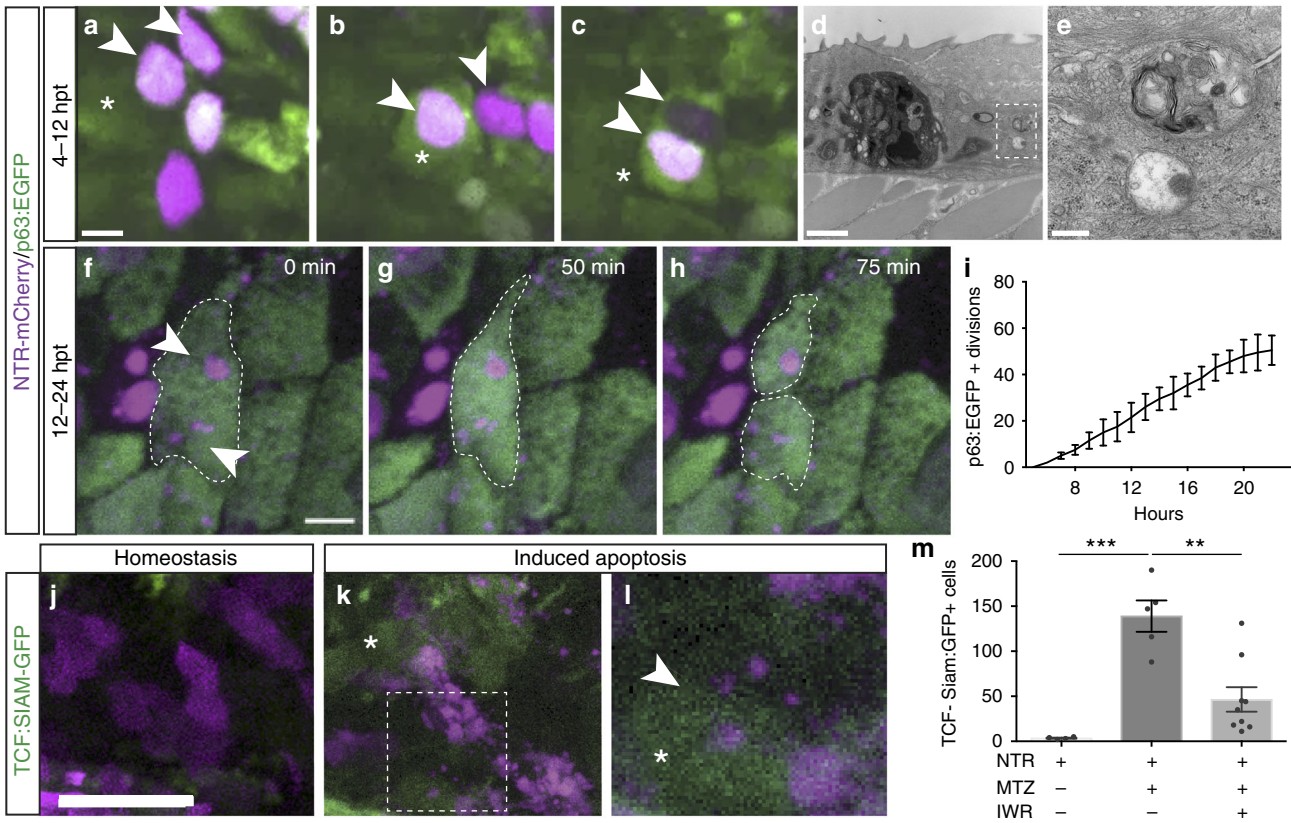

**Fig. 3** Apoptotic body uptake stimulates Wnt signaling and stem cell division. **a–c** Confocal maximum intensity projections from time-lapse imaging of a p63-positive cell (green), asterisk, engulfing two apoptotic bodies (magenta), arrowheads (scale = 5 μm), see Supplementary Movie 2. **d, e** Transmission electron micrographs of apoptotic bodies in adjacent basal cells (scale = 2 μm, 500 nm). **f–h** Time-lapse imaging of a dividing p63:EGFP-positive stem cell that has engulfed apoptotic bodies (arrowheads; scale = 5 μm), see Supplementary Movie 3. **i** Quantitation of actively dividing p63:EGFP-positive epithelial stem cells over time after induced apoptosis. **j–l** Confocal images of Wnt-responsive cells (TCF-Siam:GFP positive) after apoptosis. Arrowheads denote increased Wnt activity in a cell engulfing apoptotic bodies (scale = 50 μm). **m** Mean number of TCF-Siam:GFP-positive cells from individual larvae after stem cell ablation (n = 5) and treatment with the Wnt inhibitor IWR-1 (n = 9). Data are from three independent experiments and error bars represent sem; ***p < 0.0001, **p < 0.0007. One-way analysis of variance (ANOVA) with Dunnett's multiple comparisons test (**m**)

organelles and apoptotic material in stem cells adjacent the dying cell (Fig. 3d, e). These observations are consistent with evidence that epithelial cells can internalize living cells[51], and basal epithelial stem cells can act as non-professional, broad-specificity phagocytes to remove apoptotic cellular debris[52–54]. Fluorescently labeled macrophages $(Tg(mpeg1:EGFP)^{gl22})$ [55] also showed increased surveillance in the epithelium after induced p63-positive stem cell damage (Supplementary Fig. 4d-f), yet resulted in the engulfment of apoptotic cell corpses that averaged 8.63 μm ± 0.82 in size, significantly larger than the apoptotic bodies. Phagocytosis of apoptotic bodies can promote cell survival[56] and differentiation[49] in cultured cells, and therefore we next examined the fate of individual basal epithelial stem cells after engulfment of apoptotic bodies.

We observe a steady accumulation of proliferating p63-positive basal cells over 20 h after induced apoptosis (average of 2.8 divisions per hour), and found that 63.3 ± 4.7% of the individual dividing p63-positive stem cells contained apoptotic bodies (Fig. 3f–i and Supplementary Movie 3). Given the engulfed apoptotic bodies contain Wnt8a, we next tested if Wnt signaling is activated in the engulfing stem cell using a fluorescent biosensor $(Tg(7xTCF-Xla.Siam:GFP)^{ia4}$ [57]. p63-positive stem cells with active Wnt signaling (TCF-Siam:GFP positive) were rarely observed under homeostatic conditions in the epidermis of 4 dpf larvae (Fig. 3j). In contrast, a significant increase (38.7-fold) in the number of Wnt-responsive cells was found after induction

of damage, and those cells routinely contained apoptotic bodies (Fig. 3k, l). Pharmacological inhibition of Wnt signaling, using the tankyrase inhibitor IWR-1[58], significantly reduced (66.6%) the number of Wnt-responsive stem cells after induction of apoptosis (Fig. 3m). Together, these results suggest the stem cells that engulf Wnt-positive ESABs go on to activate Wnt signaling and divide.

**ESAB-mediated Wnt signaling regulates stem cell division**. Next, we tested if pharmacological and genetic perturbation of the Wnt pathway could prevent the apoptosis-induced proliferation. Induced expression of Dkk1, a potent inhibitor of Wnt signaling $(Tg(hsp70l:dkk1-GFP)^{w32})$[59], or treatment with IWR, resulted in a substantial decrease (35.29% and 61.31%, respectively) in the number of bromodeoxyuridine (BrdU)-positive cells after induction of apoptosis (Supplementary Fig. 4g). Further, specific perturbation of Wnt8a using CRISPR/Cas9 (clustered regularly interspaced short palindromic repeats/CRISPR-associated protein 9) genome editing significantly attenuated the apoptosis-induced proliferation (Fig. 4a–e). To confirm that Wnt8a had been altered after injection of guide RNAs (gRNA) and Cas9 protein, we analyzed the number of insertions, deletions and wild-type alleles at the Wnt8a locus in individual larvae from both injected and uninjected controls. We observe a significant increase in InDel frequencies of the sequenced alleles in injected animals (average

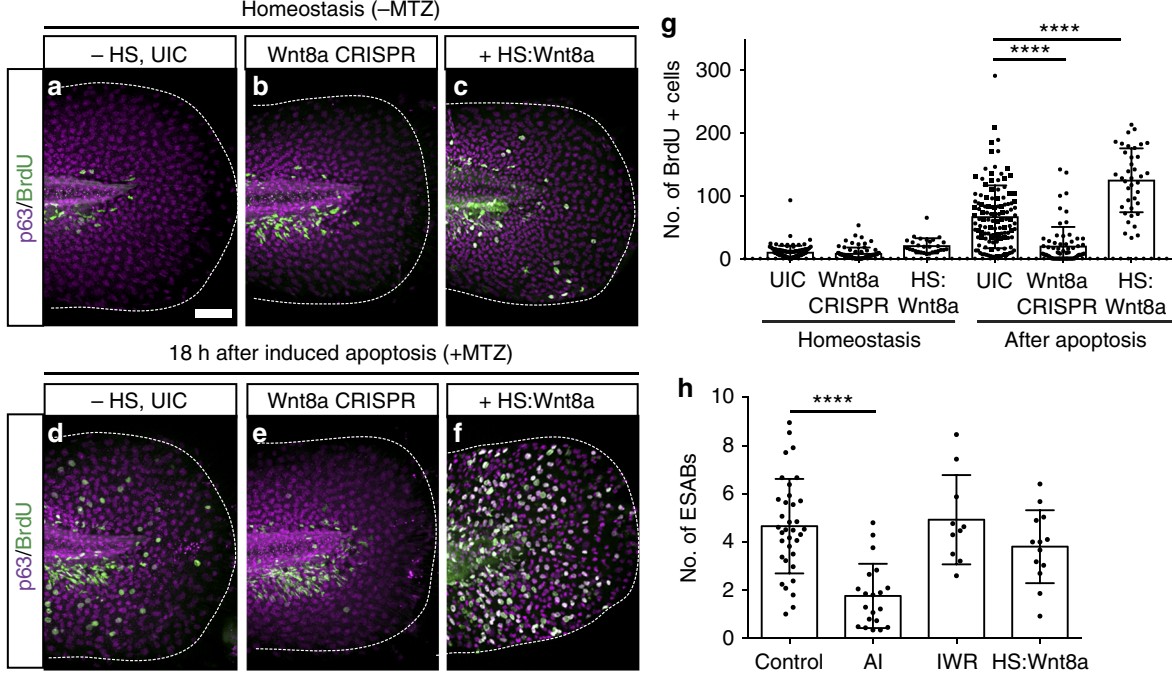

**Fig. 4** ESAB (epithelial stem cell-derived apoptotic body)-delivered Wnt8a is required for apoptosis-induced stem cell division. **a–f** Maximum intensity projections of confocal images for bromodeoxyuridine (BrdU) (green) and p63 (magenta) after perturbation of Wnt8a signaling (**a–c**), and in combination with induction of apoptosis (**d–f**) (scale = 50 μm). **g** Mean number of BrdU-positive cells from individual larvae after perturbation of Wnt8a ($n = 9$) and in combination with induced apoptosis ($n = 10$). **h** Quantitative analysis of ESAB production from controls ($n = 35$) and after treatment with apoptosis inhibitor (AI) ($n = 20$), the Wnt inhibitor IWR ($n = 10$) or overexpression of Wnt8a ($n = 13$). Data are represented as mean ± sd; ****$p < 0.0001$. One-way analysis of variance (ANOVA) with Dunnett's mutiple comparisons test (**g**, **h**)

64.97 ± 2.43), with deletions ranging from 1 to 30 bp, confirming alteration of the Wnt8a locus (Supplementary Fig. 5a-d). Further, a significant decrease in the number of Wnt8a-positive apoptotic cells, as well as the amount of detectable Wnt8a per apoptotic cell, was observed in injected animals (Supplementary Fig. 5e-i). These results suggest that apoptosis-induced Wnt8a is required to stimulate proliferation of adjacent stem cells.

To determine if overexpression of Wnt8a is sufficient to promote apoptosis-induced epithelial stem cell proliferation, we temporally induced fluorescently tagged Wnt8a ((Tg(hsp70l: wnt8a-GFP)^w34 [60]) prior to induction of apoptosis and quantified the number of dividing cells. Larvae with increased Wnt8a but not treated with MTZ showed no appreciable difference in the number of proliferating epithelial stem cells (Fig. 4a–c). Surprisingly, a significant increase in proliferation (39.72%) occurred only when increased Wnt8a was combined with induced apoptosis (Fig. 4d–g). The apoptotic bodies contained significantly increased amounts (39.53%) of fluorescently tagged Wnt8a (Fig. 5a–d and Supplementary Movie 4), while no change in the number of caspase-positive cells (Supplementary Fig. 4h) or ESABs was observed (Fig. 4h). These data support our conclusion that apoptotic-derived bodies serve as vehicles to transfer bioactive molecules, such as Wnt8a, to adjacent stem cells and promote proliferation.

To investigate if Wnt8a was present on the surface of the apoptotic bodies, we purified ESABs after induced apoptosis using differential centrifugation and characterized the $14,000 \times g$ pellet (p14) by fluorescent microscopy and flow cytometry (Fig. 5e). We found that this fraction contained 1–5 μm vesicular structures exhibiting mCherry fluorescence (Fig. 5f–j and Supplementary Figure 6a, b). These data suggest the purified fraction is significantly enriched with epithelial stem cell-derived apoptotic bodies. Immunogold labeling for Wnt8a on whole-mount purified ESABs revealed localization of Wnt8a on the

surface (Fig. 5k), while isolation of purified ESABs from Wnt8a CRISPR-injected larvae showed a significant depletion of detectable Wnt8a on the surface (Fig. 5l–n). We also detected annexin V both on apoptotic epithelial stem cells in vivo and on the surface of the purified ESABs, suggesting p63-positive stem cells externalize phosphatidylserine to the outer leaflet of the plasma membrane during apoptosis (Supplementary Fig. 6c-h). Together, these data suggest Wnt8a on epithelial stem cell-derived apoptotic bodies is key for mediating apoptosis-induced proliferation.

## Discussion

Homeostatic maintenance of epithelial tissues requires the continual removal and replacement of defective stem cells. Though apoptosis and cell division have been studied extensively, it is still not well understood how these processes are coordinated in living tissues. By combining our controlled mosaic stem cell ablation assay with in vivo time-lapse imaging to visualize individual cellular behaviors and population dynamics, we uncovered the generation of apoptotic bodies promotes both apoptotic cell clearance and mediates intercellular communication with neighboring stem cells. Our results show healthy p63-positive stem cells engulf the apoptotic bodies, activate Wnt signaling and go on to divide in an apoptosis-induced Wnt-dependent manner to restore homeostatic cell numbers (Fig. 6). We conclude that transfer of Wnt signals by apoptotic bodies stimulates proliferation of healthy stem cell neighbors to control the maintenance and repair of epithelial tissues. Given these cellular behaviors are conserved in many different types of epithelia, these studies are likely to provide valuable insights into the mechanisms regulating stem cells.

The production and delivery of morphogens such as Wnt after damage is key to stimulating localized stem cell proliferation for tissue repair. Our data suggest that dying epithelial stem cells

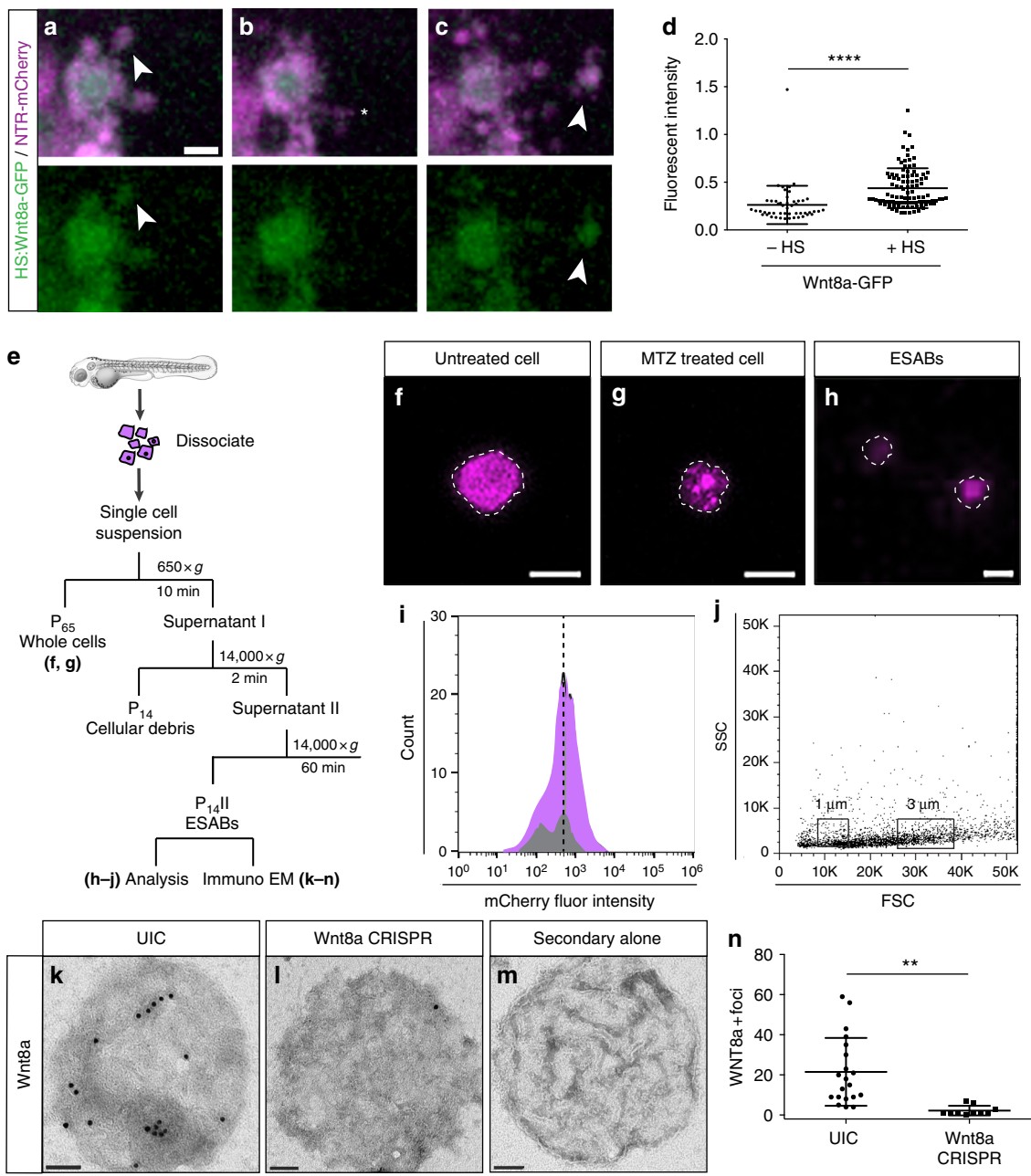

**Fig. 5** Apoptosis-induced stem cell division requires Wnt8a on the surface of ESABs (epithelial stem cell-derived apoptotic bodies). **a–c** Time-lapse confocal images of an epithelial stem cell producing an apoptotic body containing Wnt8a-GFP (scale = 5 μm), see Supplementary Movie 4. **d** Quantitation of green fluorescent protein (GFP) fluorescence in ESABs after overexpression of Wnt8a by heat-shock induction. **e** Schematic of strategy to isolate ESABs by differential centrifugation. **f–h** Whole cells (scale = 10 μm) and ESABs (scale = 5 μm) isolated by differential centrifugation exhibit mCherry fluorescence (magenta) and are appropriate size and morphology. **i** Quantification of the number of mCherry-positive ESABs after purification (magenta), compared to extracellular vesicles isolated from zebrafish larvae under homeostatic conditions (gray), using flow cytometry. **j** Size distribution of apoptotic bodies compared to size match bead controls (1 μm and 3 μm boxes, FSC = forward scatter, SSC = side scatter, see gating strategy in Supplementary Figure 6 a-b). **k–m** Transmission electron micrographs of Wnt8a immunogold labeling on whole-mount purified apoptotic bodies (scale = 500 nm). **n** The mean number of Wnt8a foci on individual ESABs from uninjected ($n = 21$) and Wnt8a CRISPR (clustered regularly interspaced short palindromic repeats)-injected ($n = 10$) animals. Data are from three independent experiments and error bars represent sd; ****$p < 0.0001$, **$p < 0.001$. Unpaired two-tailed $t$-test (**d**, **n**)

produce apoptotic bodies containing Wnt8a. Cleavage and activation of caspase-3 during apoptosis can influence over 280 downstream targets[61], and the resulting unique cytokines, microRNAs and mitogenic proteins induced by damage are packaged into apoptotic bodies and then delivered to neighboring cells. This represents a unique mechanism for generating Wnt-containing extracellular vesicles after damage that is distinct from

the formation of argosomes[62] and exosomes[63]. Thus, the molecules incorporated into epithelial stem cell-derived apoptotic bodies and the cellular mechanisms of uptake described here represent new avenues to regulate the mobilization of stem cells during tissue repair.

The inability to follow apoptotic bodies in real time has limited previous attempts to assess their function in living tissues. We

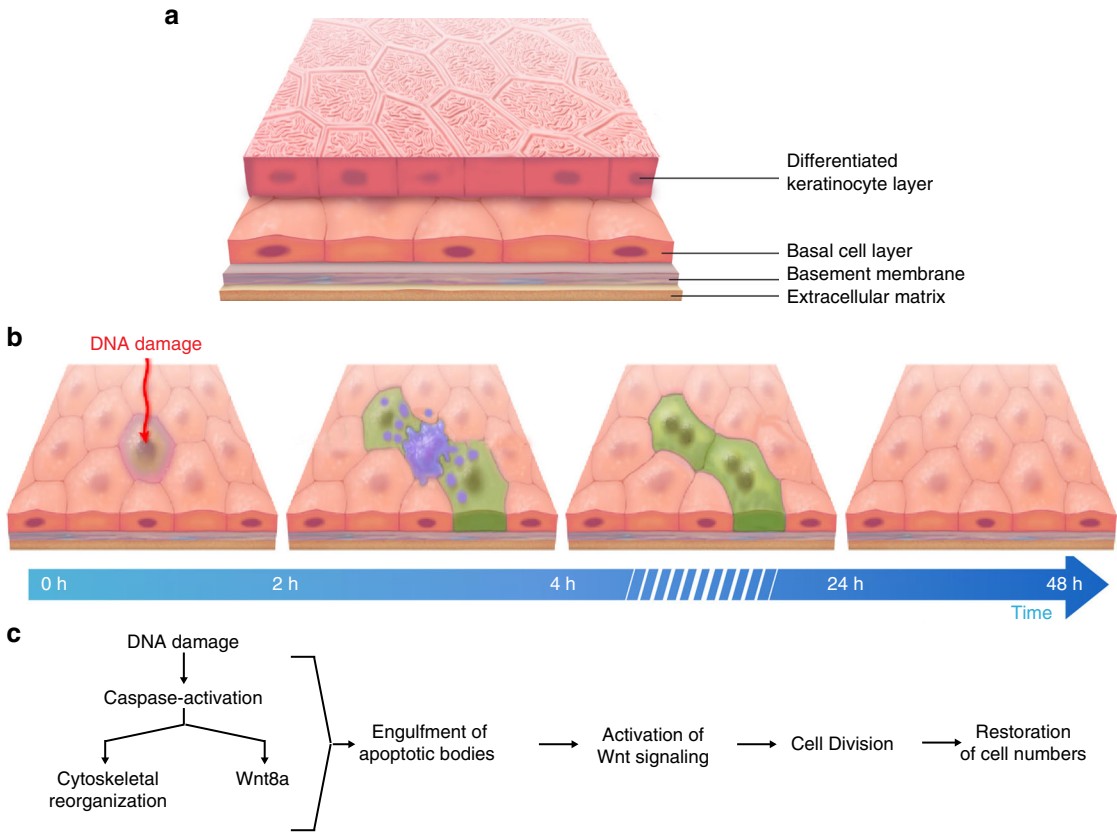

**Fig. 6** Model of apoptosis-induced stem cell turnover in an epithelial bilayer. **a** Schematic of an epithelial bilayer. **b** Sequence of cellular events observed by time-lapse imaging. **c** Temporal activation of distinct molecular mechanisms promoting the replacement of lost cells

demonstrate that engulfment of Wnt-containing apoptotic bodies by adjacent p63-positive stem cells activates Wnt signaling, consistent with uptake of Wnt-containing exosomes[63] and liposomes[64], and stimulates proliferation to sustain population numbers in the tissue. These data suggest that apoptotic body engulfment may regulate delivery of mitogenic signals to healthy stem cells and provide a way to tune the amount of localized proliferation according to the number of dying cells. The apoptotic body-based mechanism used for stem cell renewal and repair could also contribute to carcinogenesis. Engulfment of apoptotic bodies derived from dying cells associated with chemotherapy treatment could promote cancer stem cell proliferation and tumor repopulation[65,66]. In sum, the in vivo cellular and molecular characterization of epidermal stem cells reported here provides insight into engulfment of apoptotic bodies as a regulatory mechanism linking death and division to maintain overall stem cell numbers, and thus is key for epithelial tissue homeostasis.

## Methods

**Zebrafish**. Zebrafish were maintained under standard laboratory conditions with a cycle of 14 h of light and 10 h of darkness. Embryos were collected and kept in E3 embryo medium at 28.5 °C and staged as described in ref. [22]. The zebrafish used in this study were handled in accordance with the guidelines of the University of Texas MD Anderson Cancer Center Institutional Animal Care and Use Committee.

**Controlled ablation of p63-positive epithelial stem cells**. The GAL4 enhancer trap line $Et(Gal4-VP16)^{zc1036a}$, referred to as "BASAL-GET", was used to drive expression of $Tg(UAS-E1b:nsfB-mCherry)^{40}$ or $Tg(UAS-E1b:Lifeact-EGFP)$. The $Et(Gal4-VP16)^{zc1036a};Tg(UAS-E1b:nsfB-mCherry)$ line was also used in combination with $Tg(p63:EGFP)^{35}$. MTZ (Sigma, M3761) was made fresh for each assay at a 1 M concentration diluted in dimethyl sulfoxide (DMSO). The 4 dpf larvae were treated with 10 mM MTZ for 4–5 h and then either imaged, preserved for further analyses or left for 20 h to recover in fresh E3.

**Pharmacological treatments during the CAPEC assay**. The chemical apoptosis inhibitor NS3694 (Calbiochem, 178494) was diluted in DMSO as a 10 mM stock and was added to embryos at a concentration of 10 μM. The peptide caspase-3 inhibitor (Calbiochem, 264155) was diluted in DMSO as a 10 mM stock and was added to embryos at a concentration of 150 μM. Embryos were treated with either pharmacological agent for approximately 18 h prior to and during MTZ treatment. Immediately preceding MTZ treatment, fresh inhibitor and E3 were added. Inhibitor was removed for recovery period of the CAPEC assay. The chemical Wnt inhibitor IWR (Sigma, I0161) was diluted in DMSO as a 10 mM stock and was used at a concentration of 10 μM. Embryos were treated with the inhibitor for 2 h prior to MTZ treatment, during treatment and during recovery. Fresh inhibitor was added after MTZ washout during recovery.

**Temporal perturbation of Wnt signaling during the CAPEC assay**. At 18 h prior to the CAPEC assay, 3 dpf $Tg(hsp70l:dkk1-GFP)^{w32})^{59}$ or $Tg(hsp70l:wnt8a-GFP)^{w34}$ [60] embryos were placed into pre-warmed E3 media and incubated at 37 °C for 1 h. After 1 h, embryos were given fresh media and returned to 28.5 °C.

**Embryo fixation and immunofluorescence**. Zebrafish embryos were fixed for 4 h at room temperature or overnight at 4 °C with 0.4% paraformaldehyde (4% PFA), rinsed with phosphate-buffered saline (PBS), washed in PBS+0.5% Triton X-100 (PBSTX 0.5%) and blocked for 1–2 h at room temperature in 1% DMSO+2 mg/mL bovine serum albumin (BSA)+PBSTX 0.5%+10% heat inactivated goat serum. Specimens were incubated at room temperature for 4 h or overnight at 4 °C with primary antibodies in block solution. Samples were washed in PBSTX 0.5% and incubated overnight at 4 °C with secondary antibodies in block solution, rinsed with PBSTX 0.5%, and incubated with 4′,6-diamidino-2-phenylindole (DAPI) in PBSTX 0.5% for 30 min. Samples were prepped and the tail was mounted on microscope slides using glycerol with 22 × 22 coverslips.

**Primary antibodies and fluorescent dyes**. Activate caspase-3 (BD Biosciences, 559565, 1:700); BrdU (Abcam, ab6326, 1:100); Phospho-Histone H3 (H3P) (Abcam, 5176, 1:500); Tp63 (Abcam, 111449, 1:500); Tp63 (Genetex, GTX124660, 1:500); Wnt8a (ABGENT, AP21770c, 1:200); Phospho-Histone H2A.X (Ser319) (Millipore, 05–636, 1:200); Annexin V (Abcam, ab14196, 1:200); and Nucview 488 Caspase-3 Substrate (Biotium, 30029, 10 μM) were used.

**Time-lapse confocal microscopy.** Zebrafish were anesthetized with 0.04% tricaine in E3 and mounted in 1% low melt agarose in a 10 mm MatTek culture dish as in ref. [42]. The mounted specimen was imaged using a Nikon spinning disc confocal microscope or a Zeiss LSM 800 laser scanning confocal microscope.

**Scanning electron microscopy.** Samples were fixed in 3% glutaraldehyde+2% paraformaldehyde in 0.1 M cacodylate buffer (pH 7.3). Samples were washed with 0.1 M cacodylate buffer (pH 7.3), post-fixed with 1% cacodylate buffered osmium tetroxide, washed with 0.1 M cacodylate buffer and then in distilled water. Samples were treated with Millipore-filtered 1% aqueous tannic acid, washed in distilled water, treated with Millipore-filtered 1% aqueous uranyl acetate and then rinsed with distilled water. They were dehydrated with a series of increasing concentrations of ethanol and transferred to increasing concentrations of hexamethyldisilazane (HMDS) and air-dried overnight. Samples were mounted onto double-stick carbon tabs (Ted Pella. Inc., Redding, CA), and mounted onto glass microscope slides. Samples were coated under vacuum using a Balzer MED 010 evaporator (Technotrade International, Manchester, NH) with platinum alloy for a thickness of 25 nm and then flash carbon coated under vacuum. Samples were then transferred to a desiccator until examination and imaging in a JSM-5910 scanning electron microscope (JEOL, USA, Inc., Peabody, MA) at an accelerating voltage of 5 kV.

**Transmission electron microscopy.** Samples were fixed in 3% glutaraldehyde +2% paraformaldehyde+0.1 M sodium cacodylate buffer and treated with 0.1% Millipore-filtered cacodylate buffer tannic acid. They were post-fixed with 1% buffered osmium tetroxide for 30 min and stained in block with 1% Millipore-filtered uranyl acetate. Samples were dehydrated in increasing concentrations of ethanol, infiltrated and embedded in LX-112 medium. Samples were polymerized in an oven at 60 °C for about 3 days. Ultrathin sections were cut in a Leica Ultracut microtome (Leica, Deerfield, IL) and stained with uranyl acetate and lead citrate in a Leica EM Stainer. They were examined in a JEM-1010 transmission electron microscope (JEOL, USA, Inc., Peabody, MA) at an accelerating voltage of 80 kV. Digital images were obtained using AMT Imaging system (Advanced Microscopy Techniques Corp., Danvers, MA).

**Quantification of cell death and division in the zebrafish larval epithelium.** Maximum intensity projections were generated from confocal images taken at ×10 or ×20 magnification on a Zeiss LSM 800 Laser Scaning Confocal Microscope. The number of damaged (H2AX), dying (activated caspase-3) or proliferating (phosphohistone H3 (H3P) and bromodeoxyuridine labeling (BrdU)) cells were quantified in the tail fin epithelium from the posterior end to the urogenital opening as in Fig. 1a. The cells over the notochord or cells in contact with the pigment cells were excluded. Statistical analysis and graphing was performed using GraphPad Prism 6.

**Gene expression analysis.** Real-time quantitative reverse transcription PCR (qRT-PCR) was performed on total RNA isolated from 12 to 25 embryos homogenized in TRIzol (Ambion). RNA isolation was performed using RNeasy Mini Kit from QIAGEN and 1 μg to 0.5 μg RNA was used to make complementary DNA using the SuperScript III (Invitrogen) kit. qRT-PCR was done under standard PCR conditions using FastStart Essential DNA Green Master (Roche) in a LightCycler 96 (Roche) instrument. Gene expression levels were standardized to β-actin[67,68]. Statistical analysis was performed using the $2^{-\Delta\Delta CT}$[69] and Student's $t$-test. Previously published primer sequences for Wnt2b, Wnt5b, Wnt11[70], Wnt4b[71] and Axin2[72] were used. Wnt3a, Wnt8a and Lrp6 were designed using Universal Probe Library for Zebrafish (Roche).

**Primer sequences.** Wnt2bF-CCGTGAAGCAGCATTTGTCTATG
Wnt2bR-AACTTGCACTCGCATTTGGTCAT
Wnt5bF-TCAAAGAGTGCCAGTATCAGTTCAGA
Wnt5bR-GGCCACATTCGCTAGATTATACACC
Wnt11F-TCTAAACAGCAAAAGAGCGACAT
Wnt11R-GGACGAGCAGTTCCAGCGCATAT
Wnt4bF- TGCAGCAGGGGTGAACTGGA
Wnt4bR- TGCCTTTCGCTCTTTCCGGC
Axin2F- CCAGCAGCAAAGCCTTCAGT
Axin2R- GCGCGCACAAAGTAGACGTA
Wnt3aF-CGTGTCATGGCAAGCTACC
Wnt3aR-CTGCGTACCCAGAGAGGTGT
Wnt8aF- GGAAAGCGCACTGCAGTTAT
Wnt8aR-AACTCCAGCACTTATAGCA
Lrp6F-AGACGAGATCGGCTGCTATG
Lrp6R-TTGAGCCGATGGTGTTAGTG.

**Perturbation of Wnt8a using CRISPR/Cas9 genome editing.** Chop-Chop (https://chopchop.rc.fas.harvard.edu) was used to identify target sites and gRNA sequences within the zebrafish Wnt8a (ENSDARG00000052910, v10). We injected

1 nL of gRNAs targeting exon 2 (TTGCATCTCAAGAAGAAGGG) and exon 5 (ATGGGGGACTTCGAAAACTG) with 2 μm of nls-Cas9-nls (NEB, cat. no. M0641T). Microinjections of guide RNAs for both exon 2 and 5 animals, or each guide independently, yielded similar results. To verify editing at the desired location, DNA was isolated from individual embryos as in ref. [73] and amplified using primers specific to the target site: (Exon2F-TTGAAGTGCAATTAAGGCAGAA, Exon2R- ATAACCAACGACGCAAAAATCT, Exon5F-TTGAAGTGCAAT-TAAGGCAGAA, Exon5R-ATAACCAACGACGCAAAAATCT). Deep sequencing (described below) was performed to determine the number and type of insertions or deletions at the target site.

**Genotyping Wnt8a CRISPR-injected larvae by bar-coded deep sequencing.** To accurately correlate between phenotype and genotype in individual CRISPR-injected animals, tails of individual 4 dpf animals were isolated for immunohistochemistry experiments and the remaining tissue from each animal was then utilized for DNA extraction and genotyping as described in ref. [73]. Then, 1 μL of extracted DNA was used to PCR amplify the CRISPR-targeted regions using a three primer strategy that adds a unique 6 bp barcode to each sample to facilitate discrimination of individual animals as described by ref. [74]. Next, 6 μL of each uniquely bar-coded PCR reaction was pooled, run through a MinElute Reaction Cleanup Kit (Qiagen, cat. no. 28204) and deep sequencing was performed on a MiSeq instrument using a V2, 500 cycle kit. The resulting reads were demultiplexed using the FASTQ/A Barcode Splitter tool from the FASTX-Toolkit to identify and isolate the barcodes assigned to individual animals. To estimate the amount of CRISPR-mediated editing, fastq files for individual animals were run though the command line version of the CRISPResso software pipeline in the amplicon mode[75]. To account for sequencing errors and ensure proper identification of edited vs non-edited animals, we set a cutoff of a minimum of 40% editing required to be considered mutant, and removed any uninjected animals that displayed greater than 10% editing from the analysis. Targeted bar-coded deep sequencing revealed similar indel frequencies for both exon 2 and 5.

**Tracking and quantification of macrophage behavior after induced epithelial stem cell death.** Et(Gal4-VP16)$^{zc1036a}$;Tg(UAS-E1b:nsfB-mCherry); (Tg(mpeg1: EGFP)$^{gl22}$) larvae were imaged over time under homeostatic conditions or after induced epithelial stem cell apoptosis. Fluorescently labeled macrophages were manually tracked frame by frame using Imaris (Bitplane). Quantification of apoptotic cell corpse and ESAB engulfment, as well as determination of the size of ingested material, was performed using Zeiss Zen software.

**ESAB purification by differential ultracentrifugation.** Epithelial stem cell-derived apoptotic bodies were isolated using differential centrifugation. Et(Gal4-VP16)$^{zc1036a}$;Tg(UAS-E1b:nsfB-mCherry) larvae treated with MTZ or untreated controls were both incubated in Trypsin EDTA 0.25% for 30 min. Larvae were then dissociated with a pestle and subjected to a 10 min of centrifugation at 650 × g. The supernatant was transferred and centrifuged for 2 min at 14,500 × g. The supernatant was once again transferred and centrifuged for 1 h at 14,500 × g. The pellet was washed twice and suspended in dPBS. The purified fraction containing the ESABs was characterized using flow cytometry on a Beckman Coulter Gallios, and the subsequent data were analyzed using FlowJo. Beads of known sizes 1.1 (Sigma LB11–1mL) and 3 μm (Sigma LB30–1mL) were used as standards to set gates for determining the size of the apoptotic bodies.

**Immunogold labeling and transmission electron microscopy.** Samples were fixed in either 2% paraformaldehyde or 2% glutaraldehyde in 0.1 M PBS buffer, pH 7.4. Purified apoptotic bodies were left to settle on carbon-coated grid. For immunogold labeling, pelleted apoptotic bodies were plated on grids, blocked and stained with annexin V or anti-Wnt8a antibody (1:200), then incubated in secondary Goat Anti-Rabbit 15 nm Gold (Abcam ab27236). Each staining step was followed by 5 PBS washes and finally 10 washes in H$_2$O before contrast staining with 3% uranyl acetate. After staining with 3% uranyl acetate, grids were air-dried and visualized using a transmission electron microscope (JOEL JEM-1010).

**Reporting summary.** Further information on experimental design is available in the Nature Research Reporting Summary linked to this article.

## Data availability

The source data underlying Figs. 1j, k, 2c,g, and h, 3i and l, 4g, h, 5d and n, and Supplementary Figs 1b and d, 2d and g, 3a, e and c, 4f-h, 5d and i, and 6h are provided as a Source Data file. All relevant data are available from the authors upon request. Further information and requests for resources and reagents should be directed to and will be fulfilled by the Lead Contact, George T. Eisenhoffer (gteisenhoffer@mdanderson.org).

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

## Acknowledgements

We thank members of the Arur, Galko and Behringer laboratories for scientific discussions, suggestions and comments. This work was supported by the Cancer Prevention Institute of Texas, RR14007, and National Institutes of General Medical Sciences, 1R01GM124043, to G.T.E. We thank Kenneth Dunner Jr at the UT MD Anderson High Resolution Electron Microscopy Facility for assistance with the electron microscopy data and Wendy Schober at the Flow Cytometry and Cellular Imaging Core for help with the flow cytometry. The High Resolution Electron Microscopy and Flow Cytometry and Cellular Imaging Core facilities were supported by CCSG grant NIH P30CA016672. The Sequencing and Microarray facility was supported by NCI Grant CA016672.

## Author contributions

G.T.E. designed the research. C.K.B., S.T.W., O.E.R., K.M.S., A.M. and E.A.S. performed experiments and analyzed data. C.K.B. and G.T.E. wrote the paper, and all authors provided edits.

## Additional information

**Competing interests:** The authors declare no competing interests.

