## [Peer Review File · Nature Communications]

Reviewers' comments:

Reviewer #1 (Remarks to the Author):

In this manuscript, the authors elegantly demonstrated the generation of Wnt8a-containing apoptotic bodies by apoptotic epithelial stem cells under in vivo conditions. The presence of these apoptotic bodies can subsequently activate Wnt signalling and promote cell proliferation of neighbouring stem cells. Overall, the manuscript is very novel, well written and highly relevant to the field of apoptotic cell disassembly and clearance. The experiments presented were well performed and the conclusions were largely justified. The authors should, however, consider the following points:

1. The approaches used by the authors to monitor cell death and associated morphological changes were well performed. Could the author indicate where the 'noticeable protrusions' generated from apoptotic basal stem cells is on the EM images in Supplemental Figure 1e.
2. In Figure 1k, # of BrdU+ cells were quantified, are all these cells p63+?
3. The use of NS3694 is a nice approach to inhibit apoptosis without targeting caspase activity. I think the authors should highlight this point to make it clear to the reader that caspase activity in regulating non-apoptotic processes were not affected.
4. Wnt4b was also highly unregulated under apoptosis inducing condition as shown in Supplemental Figure 3a. Can the authors please discuss the significance of this finding?
5. The authors made the conclusion that "these data indicate that Wnt8a is produced in the apoptotic cells in a caspase-dependent manner" (page 7). To better support this conclusion, caspase inhibitor could be used in experiments shown in Figure 2. Otherwise, the current data support the upregulation of Wnt8a during apoptosis.
6. Regarding the kinetics of Wnt8a upregulation, does it occur prior to apoptotic body formation? Wnt8a should be observed prior to apoptotic morphologies?
7. The authors made the comment that the protrusions generated during apoptosis were 'actin-based', assuming based on the lifeact-GFP experiment shown in Supplemental Figure 4. However, the presence of GFP in a particular morphological structure does not necessary mean the presence of F-actin or indicates this process is actin-based? Thus, it is difficult to make a strong conclusion unless cells were stained with SiR-Actin or phalloidin.
8. It is interesting to note the rapid loss of NTR-mCherry signals in apoptotic bodies in the Supplemental Video. Is this an indication of apoptotic body lysis (could be inside or outside phagocytes)?
9. Could the authors please comment on whether the engulfment step is necessary for apoptotic bodies to stimulate cell proliferation.
10. Figure 5h and 5i: Does ESABs expose phosphatidylserine? Are they positive for annexin V staining? What exactly is 'control (grey)' in Figure 5i?
11. Figure 5h and 5k: ESABs in Figure 5h appear to be 4-5 um, but ESABs in Figure 5k appear to be 500-600 nm. Could the authors please clarify the discrepancy?

Kind Regards,

Dr Ivan Poon

Reviewer #2 (Remarks to the Author):

The submitted manuscript by Brock et al. describe an interesting new finding demonstrating that apoptotic bodies shedded from dying cells are capable of inducing Wnt signaling pathways in basal cells that engulf these vesicles, leading to cell proliferation and tissue turn-over. This manuscript provides evidence that apoptotic cells trigger via a Wnt8 mechanisms cell proliferation in the surrounding cells to control epithelial over.

This is a well written and interesting study that provides a novel role for shedded vesicles of cell undergoing apoptosis to influence their surrounding. Although the study could go into more depths exploring the physiological consequences and mechanisms of Wnt8-presenting vesicles, this is an interesting finding with potentially broad implications (also in other organisms/systems) which in my view warrants publications.

I have a few points that should be addressed in a revised manuscript:

Major point:

- Conceptually, I am not fully convinced that these are a 'novel' class of vesicles and it is not clear for me how the others distinguish these vesicles from microvesicles that are shedded from the membrane and that have previously been shown to carry Wnt proteins
- Fig. 4: the authors likely mean IWR-1 (a Tankyrase inhibitor) which be properly mentioned and explained in the text - Tankyrase inhibitors are also not absolutely Wnt specific and limitations of this experiments should be discussed (p.9.).
- A more specific Wnt inhibitor which is also more relevant to the Wnt production the authors describe is LGK974 or IWP-2. The authors should repeat the experiments (also Figure 5) with a porcupine inhibitor as a control for Wnt secretion.
- The authors should test whether an increase in mRNA levels after induction of apoptosis leads to a concurrent increase in the secretion

With the above limitations, I found the manuscript overall interesting and suitable after revisions for Nature Communications.

Reviewer #3 (Remarks to the Author):

This is a very interesting and impressive manuscript using the zebrafish embryo to reveal an important mechanism of tissue homeostasis: dying basal keratinocytes express Wnt8a, which via the engulfment of the resulting apoptotic bodies is taken up by neighboring basal keratinocyte in which it promotes cell proliferation, compensating for the loss of the apoptotic cells. This mechanism, although in light of former studies not very surprising, has in its entirety never been demonstrated before. Furthermore, and most importantly, it is shown in vivo in a living epithelium. The presented data are definitively interesting and novel enough to merit publication in Nature Communications. Furthermore, the manuscript is very well written, and the data of highest technical quality. However, prior to acceptance for publication, some revisions, including the addition of more experimental data, would be helpful to further substantiate some of the most crucial observations.

Major points:

1. The authors present very convincing data that apoptotic bodies are engulfed by basal keratinocytes. However, this does not rule out the simultaneous function of other phagocytes, such as macrophages. Double stainings with lyz and/or mpeg1 (in situs or with the respective transgenic lines) could be carried out to search for the presence of Wnt8a-positive apoptotic bodies in skin macrophages. Furthermore, basal keratinocyte proliferation rates could be determined in keratinocyte-ablated larvae lacking innate immune cells (for instance obtained via pu.1 morpholino injection to block myeloid cell lineage development).
2. In the cited work by Huh et al. (2004), the effect of dying cells in *Drosophila* wing buds on proliferation on surviving neighbors is claimed to be independent of cell death, based on the finding that overexpression of the Baculovirus p35 protein, although blocking cell death, did not block the hyperproliferation of neighboring cells. This is in contrast to the reduction of both apoptosis and proliferation rates obtained here upon treatment with the apoptosome inhibitor NS3694 (Fig. 2c). Are these different outcomes due to differences in the underlying mechanisms, or due to the different reagents used to block apoptosis? To address this, the authors might want to test other caspase inhibitors (e.g. Z-VAD-FMK), to see whether they can uncouple the effects on cell death and cell proliferation.
3. NTR/MTZ-induced apoptosis might be considered as a rather artificial type of cell death. Therefore, it would be helpful to repeat at least some of the most crucial experiments for instance after UV-induced death of basal keratinocytes.

Minor points:

4. Figure legends in the main text are often not detailed enough to explain figures. For example, Figure 5 j: what do SSC and FSC stand for? In Figure 5 i: what is the control? Figure 3 k': what is the asterisk indicating?
5. Some figures are missing information that would aid understanding. Figure 3 a-h: label colours; Figure 4 a-f: label colours.
6. Figure 3 j-l: how are individual GFP+ cells counted? From the images provided, it appears that the GFP is too weak to easily identify cell borders, and nuclei are not labeled.
7. Statistical methods: the use of the T-test is inappropriate when comparing more than two groups (i.e. for quantifying BrdU- or caspase-positive cells). Use ANOVA when comparing multiple groups.
8. For the Wnt8 Crispr, the rationale behind using only the injected fish is not explained. Is it because the stable line gives a gastrulation phenotype like described Wnt8 morphants? If so, the rationale should be better explained. Furthermore, the materials and methods indicates that exons 2 and 5 were targeted using a single guide RNA for each. However, it is not clear in the main text or in the figure legends which exon was targeted in the examples shown (although Supplementary Figure 5 shows indel frequencies for exon 2 only), nor is it stated whether similar results were obtained for both exons.
9. The rationale for the choice of the Wnt genes examined by RT-PCR should be better explained.
10. Were the other upregulated Wnts (Wnt4b, Wnt11) localized to apoptotic cells or not? It is not clear in the text why Wnt4b and Wnt11 were excluded from further analysis.
11. Figure 3 d-e: Is the dying cell really a basal cell? It looks like it could be a peridermal cell.
12. The authors have previously shown that inducing apoptosis in zebrafish embryos by treatment with G418 leads to apical extrusion of peridermal cells, rather than engulfment by neighboring cells. Why is the outcome so different here? This is alluded to briefly in the results, but should be explored further in the discussion.
13. In the Discussion, the authors might want to speculate on the nature of the (cell death-dependent) cofactor, which is required by Wnt8a to exhibit its pro-proliferative gain-of-function effect.
14. Please cite the work by Fischer et al. (PLoS Genet 2014), pointing to the existence of caspase-

dependent mitogens generated by “naturally dying” outer keratinocytes of zebrafish breeding tubercles to stimulate compensatory proliferation of basal keratinocytes, assuring proper homeostasis of these self-renewing epidermal appendages.

Response to Reviewers:

We would like to thank all the reviewers for their insightful comments and constructive feedback on the new mechanism presented in our manuscript. We have addressed all of the comments from the reviewers and changes to revised manuscript have been highlighted. We have also included a summary of the changes to the manuscript at the end of this letter.

Reviewer #1 (Remarks to the Author):

In this manuscript, the authors elegantly demonstrated the generation of Wnt8a-containing apoptotic bodies by apoptotic epithelial stem cells under in vivo conditions. The presence of these apoptotic bodies can subsequently activate Wnt signalling and promote cell proliferation of neighbouring stem cells. Overall, the manuscript is very novel, well written and highly relevant to the field of apoptotic cell disassembly and clearance. The experiments presented were well performed and the conclusions were largely justified. The authors should, however, consider the following points:

1. The approaches used by the authors to monitor cell death and associated morphological changes were well performed. Could the author indicate where the ‘noticeable protrusions’ generated from apoptotic basal stem cells is on the EM images in Supplemental Figure 1e.

The “protrusions” were in reference to bulges of the surface epithelium due to the presence of the rounded up basal apoptotic cells. These data also showed that the surface periderm remains intact. We have amended the main text on page 5 to refer to these bulges within the surface epithelium and have added arrowheads to Supplementary Figure 1e to highlight the bulges when viewed from above and in cross section.

2. In Figure 1k, # of BrdU+ cells were quantified, are all these cells p63+?

We found that the large majority (97.87%) of the BrdU+ cells analyzed in Figure 1k after induced apoptosis were p63 positive, and this data has been added to the main text on page 6.

3. The use of NS3694 is a nice approach to inhibit apoptosis without targeting caspase activity. I think the authors should highlight this point to make it clear to the reader that caspase activity in regulating non-apoptotic processes were not affected.

We would like to thank the reviewer for bringing this point to our attention. We have now added a sentence on page 6 to clarify that caspase activity in regulating non-apoptotic processes was not affected using this approach. We also go on to compare this to new data specifically targeting caspase activity using the peptide inhibitor zDEVD-fmk.

4. Wnt4b was also highly unregulated under apoptosis inducing condition as shown in Supplemental Figure 3a. Can the authors please discuss the significance of this finding?

We were also intrigued by the significant increase in Wnt4b expression after induction of apoptosis, especially given that Wnt4 was recently found to be upregulated during the early

phases of cisplatin-induced acute kidney injury (He YX et al., Scientific Reports 2018). Wnt4, and Wnt11, were also found to be expressed specifically in crypts of embryonic and adult small intestine and are required for homeostasis of the intestinal epithelium (Pinto D. et al., Genes and Development 2003). While it is an intriguing target, the lack of zebrafish specific antibodies or transgenic lines to induce Wnt4b expression prevented us from following up with Wnt4b at this time, and thus, we chose to focus solely on Wnt8a for this manuscript.

5. The authors made the conclusion that ‘these data indicate that Wnt8a is produced in the apoptotic cells in a caspase-dependent manner’ (page 7). To better support this conclusion, caspase inhibitor could be used in experiments shown in Figure 2. Otherwise, the current data support the upregulation of Wnt8a during apoptosis.

To better support our conclusions, we expanded our studies to include inhibition of caspase-3 and how this impacts production of Wnt8a. We now present data that suggests inhibition of caspase-3 using zDEVD-fmk also significantly reduces both the number of caspase-3 cells (Supplemental Figure 1d) and those with apoptosis-induced Wnt8a expression (Figure 2c). These data support our conclusion Wnt8a is produced in the apoptotic cells in a caspase-dependent manner. We have updated the main text on pages 6 and 7 to reflect these changes.

6. Regarding the kinetics of Wnt8a upregulation, does it occur prior to apoptotic body formation? Wnt8a should be observed prior to apoptotic morphologies?

Our data timelapse imaging experiments to define the dynamics and kinetics of apoptotic body generation (Figure 2 d-h) and caspase-3 cleavage and activation (Supplementary Figure 3d-e) showed a significant increase of caspase-3 cleavage that corresponded with cell rounding within 70 minutes of addition of the prodrug to induce cell death. We observe a significant increase in the detectable Wnt8a-positive cells at 1 hour post-treatment (Supplementary Figure 3b-c). Increases in detectable Wnt8a appeared during cell rounding, but prior to apoptotic morphologies such as membrane blebbing or apoptotic body formation. We are currently working to further resolve the timing and regulation of these events.

7. The authors made the comment that the protrusions generated during apoptosis were ‘actin-based’, assuming based on the lifeact-GFP experiment shown in Supplemental Figure 4. However, the presence of GFP in a particular morphological structure does not necessary mean the presence of F-actin or indicates this process is actin-based? Thus, it is difficult to make a strong conclusion unless cells were stained with SiR-Actin or phalloidin.

We have successfully used phalloidin to label F-actin in dying basal stem cells, yet this approach failed to capture the rapid dynamics of cytoskeletal reorganization required for apoptotic body generation and clearance. We agree with the reviewer that the presence of GFP does not necessary mean the presence of F-actin or that the process is actin-based. To address this point, we attempted to use SiR-Actin as an independent method to examine F-actin in living basal cells. While the SiR-Actin dye worked well to label F-actin in surface epithelial cells, we were unable to get consistent labeling of the basal p63-positive stem cell population, likely due to lack of penetration. To address this point, we have changed the text on page 8 to read that “the

cellular extensions containing the F-actin marker Lifeact” and have removed the statement about the process being actin based.

8. It is interesting to note the rapid loss of NTR-mCherry signals in apoptotic bodies in the Supplemental Video. Is this an indication of apoptotic body lysis (could be inside or outside phagocytes)?

The rapid loss or decay of mCherry fluorescence is indeed quite striking. While these data suggest that the apoptotic bodies are being degraded and/or lysing, it is currently not possible to definitively track and quantify degradation or lysis without a secondary label of the plasma membrane. We are currently working to tag the apoptotic cells/bodies with an independent fluorophore that specifically labels the plasma membrane to circumvent this issue and analyze this process in future studies.

9. Could the authors please comment on whether the engulfment step is necessary for apoptotic bodies to stimulate cell proliferation.

The reviewer poses an excellent question, and one that we have been focused on trying to address. The current level of resolution in our assays using the fluorescent tcf-reporter (in Figure 3j-k) does not distinguish between cells that have engulfed ABs and those that interacted with individual ABs and activated Wnt signaling. The pharmacological inhibitors we have tested that alter cytoskeletal reorganization required for engulfment also interfere with apoptotic body production. Therefore, we have not been able to experimentally uncouple production of apoptotic bodies from engulfment, preventing such an analysis at this time.

10. Figure 5h and 5i: Does ESABs expose phosphatidylserine? Are they positive for annexin V staining? What exactly is ‘control (grey)’ in Figure 5i?

We examined localization of phosphatidylserine (PS) in apoptotic p63-positive stem cells and associated apoptotic bodies, as well as in purified ESABs. We found that *in vivo*, apoptotic epithelial stem cells externalize PS to the surface (as detected by annexin V staining), and that PS can also be detected on the purified ESABs. We also now show that purified ESABs have a large number of PS positive foci on the surface. These data have been added as Supplementary Figure 6a-f.

The control (grey area in the FACS plot in Figure 5i) is extracellular vesicles isolated from zebrafish larvae under homeostatic conditions, and therefore, not epithelial stem cell induced apoptosis. We have added a sentence to the figure legend to clarify this point.

11. Figure 5h and 5k: ESABs in Figure 5h appear to be 4-5 μm , but ESABs in Figure 5k appear to be 500-600 nm. Could the authors please clarify the discrepancy?

We would like to thank the reviewers for pointing out this discrepancy. The scale bar is actually 500nm, making the ESABs in Figure 5k approximately 1-1.2 μm , within the 1-5 μm size range defined for apoptotic bodies.

Reviewer #2 (Remarks to the Author):

The submitted manuscript by Brock et al. describe an interesting new finding demonstrating that apoptotic bodies shedded from dying cells are capable of inducing Wnt signaling pathways in basal cells that engulf these vesicles, leading to cell proliferation and tissue turn-over. This manuscript provides evidence that apoptotic cells trigger via a Wnt8 mechanisms cell proliferation in the surrounding cells to control epithelial over.

This is a well written and interesting study that provides a novel role for shedded vesicles of cell undergoing apoptosis to influence their surrounding. Although the study could go into more depths exploring the physiological consequences and mechanisms of Wnt8-presenting vesicles, this is an interesting finding with potentially broad implications (also in other organisms/systems) which in my view warrants publications.

I have a few points that should be addressed in a revised manuscript:

Major point:

- Conceptually, I am not fully convinced that these are a 'novel' class of vesicles and it is not clear for me how the others distinguish these vesicles from microvesicles that are shedded from the membrane and that have previously been shown to carry Wnt proteins

To our knowledge, little is known is about the production of apoptotic bodies from epithelial stem cells in living tissues. Recent studies have shown that damaged cells activate specific genetic programs aimed at survival and proliferation (Sun, G et al., Journal Cell Biology 2017). As the cell proceeds toward death and not repair, it is not well understood if these unique contents are shuttled to neighboring cells. Our data suggests that pro-mitogenic cues and other factors are packaged into the apoptotic bodies for transfer to neighboring cells. We have now been able to assign function to these extracellular vesicles generated only by apoptotic epithelial stem cells by following their fate *in vivo*. While healthy cells produce microvesicles that have been shown to carry Wnt, such as argosomes (Greco, V., Cell 2001) and exosomes (Gross, J.C., Nat Cell Bio 2012), the ESABs described here are larger and are derived from apoptotic epithelial stem cells. Our hope is that addressing the critiques presented here will help convey the novelty to the reviewer.

- Fig. 4: the authors likely mean IWR-1 (a Tankyrase inhibitor) which be properly mentioned and explained in the text - Tankyrase inhibitors are also not absolutely Wnt specific and limitations of this experiments should be discussed (p.9).

The authors thank the reviewer for bringing this point to our attention. To further clarify this point, we have modified the sentence in the main text that describes IWR-1 as a Tankyrase inhibitor. To address the limitations of specificity using pharmacological inhibition, our study includes independent tests to block all Wnt-Fz-LRP5/6 interactions by induced expression of the genetically-encoded Dkk1 inhibitor (Supplemental Figure 4g).

- A more specific Wnt inhibitor which is also more relevant to the Wnt production the authors describe is LGK974 or IWP-2. The authors should repeat the experiments (also Figure 5) with a porcupine inhibitor as a control for Wnt secretion.

We attempted to perturb Wnt production and secretion using the suggested pharmacological inhibitors LGK974 and IWP-2 at concentrations previously reported for use in early stage zebrafish embryos (Xang, X, J Med Chem 2014). Unfortunately, the results obtained with LGK974 were highly variable because the inhibitor periodically dropped out of solution, likely due to incompatibility with MTZ required for the inducible apoptosis assay. Likewise, treatment with the porcupine inhibitors, IWP-2 and IWP-12, resulted in significant mortality, even with brief treatment periods. The high mortality rate seen after treatment with these inhibitors was not observed with IWR-1 or genetic inhibition of Wnt signaling, and therefore, we did not feel confident moving forward with this approach at this time. To our knowledge, these compounds have not been used on four day old post-fertilization zebrafish larvae, and therefore, require more optimization before being useful to perturb Wnt secretion in this context.

- The authors should test whether an increase in mRNA levels after induction of apoptosis leads to a concurrent increase in the secretion

Our initial experiments to induce production of fluorescently-tagged Wnt8a by heat shock under homeostatic conditions showed no increase in proliferation above baseline levels without damage (Figure 4c, f & g). We hypothesized that if Wnt8a is secreted, this should lead to an increase in proliferation in neighboring cells. Yet, this is not what we observe. We were able to observe fluorescently-tagged Wnt8a in and on the surface of the apoptotic cell corpse and bodies (Figure 5a-c and k-n). We did not, however, detect any fluorescently-tagged Wnt8a outside of the apoptotic cell. Given the *in vivo* nature of these experiments, we do not have the resolution with the current tools to determine if the newly synthesized Wnt8a is directly leading to increased Wnt8a secretion. We are working on both pharmacological and genetic based approaches to perturb apoptotic body generation or uptake by neighboring cells that will give us the resolution to address these key questions moving forward.

Reviewer #3 (Remarks to the Author):

This is a very interesting and impressive manuscript using the zebrafish embryo to reveal an important mechanism of tissue homeostasis: dying basal keratinocytes express Wnt8a, which via the engulfment of the resulting apoptotic bodies is taken up by neighboring basal keratinocyte in which it promotes cell proliferation, compensating for the loss of the apoptotic cells. This mechanism, although in light of former studies not very surprising, has in its entirety never been demonstrated before. Furthermore, and most importantly, it is shown in vivo in a living epithelium. The presented data are definitively interesting and novel enough to merit publication in Nature Communications. Furthermore, the manuscript is very well written, and the data of highest technical quality. However, prior to acceptance for publication, some revisions, including the addition of more experimental data, would be helpful to further substantiate some of the most crucial observations.

With the above limitations, I found the manuscript overall interesting and suitable after revisions for Nature Communications.

Major points:

1. The authors present very convincing data that apoptotic bodies are engulfed by basal keratinocytes. However, this does not rule out the simultaneous function of other phagocytes, such as macrophages. Double stainings with lyz and/or mpeg1 (in situ or with the respective transgenic lines) could be carried out to search for the presence of Wnt8a-positive apoptotic bodies in skin macrophages. Furthermore, basal keratinocyte proliferation rates could be determined in keratinocyte-ablated larvae lacking innate immune cells (for instance obtained via pu.1 morpholino injection to block myeloid cell lineage development).

We would like to thank the reviewer for bringing up this important point about the simultaneous function of the immune system in this process. Rationalizing that these interactions would be highly dynamic and may not be truly reflected by static analyses, we used timelapse confocal microscopy of mpeg1:EGFP transgenic animals with the macrophages fluorescently labeled to determine their contribution to clearance of the apoptotic cells and bodies from the epithelial tissue. We observed an increased number of circulating macrophages within the epithelial tissue after induced apoptosis (Supplementary Figure 4d-f). We tracked the migration and path of these cells as they traveled through the tissue after induced death, and found that the macrophages surveyed the area around clusters of apoptotic cells. We also found that the circulating macrophages engulfed apoptotic cell corpses that averaged $8.629 \pm 0.82 \mu\text{m}$ in size, much larger than apoptotic bodies. Surprisingly, areas of the tissue containing only apoptotic bodies did not show significant recruitment of the macrophages. Our data suggest macrophages only account for a small number of total clearance events and do not appear to specifically target the apoptotic bodies for engulfment. Together, these data support our conclusion that neighboring epithelial stem cells are the primary mode of clearance of the epithelial stem cell derived apoptotic bodies.

Our system now sets up a novel platform to address the functional significance of the immune system after epithelial stem cell damage, yet this characterization will require further experimentation that is beyond the scope of this current manuscript. We were also concerned that knockdown of PU.1 using antisense morpholino oligonucleotides to disrupt myeloid lineage production may not persist until 4 days post-fertilization when we perform our assay (as has been reported for other morpholino based approaches). We have instead begun using a novel mpeg1::NTR-mCherry transgenic line from the Wagner laboratory at Rice University crossed to our zc1036a:GAL4;UAS:NTR-mCherry enhancer trap line to genetically ablate the macrophages at the same time as the epithelial stem cells. Our preliminary results using this approach suggest there is no change in the apoptosis-induced proliferation in the presence or absence of the macrophages. We are also independently pursuing CRISPR mediated perturbation of the myeloid lineage and plan to use this approach in future studies.

2. In the cited work by Huh et al. (2004), the effect of dying cells in Drosophila wing buds on proliferation on surviving neighbors is claimed to be independent of cell death, based on the finding that overexpression of the Baculovirus p35 protein, although blocking cell death, did not

block the hyperproliferation of neighboring cells. This is in contrast to the reduction of both apoptosis and proliferation rates obtained here upon treatment with the apoptosome inhibitor NS3694 (Fig. 2c). Are these different outcomes due to differences in the underlying mechanisms, or due to the different reagents used to block apoptosis? To address this, the authors might want to test other caspase inhibitors (e.g. Z-VAD-FMK), to see whether they can uncouple the effects on cell death and cell proliferation.

To explore alternative ways to inhibit cell death in our system, we tested the ability of both zVAD-fmk and zDEVD-fmk to block cell death and inhibit proliferation in our system. We found that larvae treated with zDEVD-fmk had decreased numbers of caspase positive cells (Supplementary Figure 1d) that corresponded with a significant reduction in the amount of apoptosis-induced proliferation (Figure 1k). See also Reviewer #1 Point 5. Our pharmacological approach does not currently allow us to uncouple the effects of cell death and cell proliferation. We are actively working to develop genetic tools that would allow for inducible caspase expression and/or factors that enhance survival to test these ideas. To our knowledge, Hu et al. did not explore if the “undead” cells were blebbing and generating apoptotic bodies after damage. Thus, at this time, it is difficult to determine if the differences lie in the approach or underlying biological mechanism.

3. NTR/MTZ-induced apoptosis might be considered as a rather artificial type of cell death. Therefore, it would be helpful to repeat at least some of the most crucial experiments for instance after UV-induced death of basal keratinocytes.

We had been independently exploring UV-induced epidermal cell death for another ongoing project in the laboratory investigating how damaged cells are eliminated after amputation. These studies showed that exposure to UV damages both the periderm cells and basal keratinocytes at the same time, and that cleaved caspase-3 is detected within four hours of UV-exposure. Under these conditions, the damaged periderm cells are eliminated by extrusion, consistent with our previous reports (Eisenhoffer et al., 2012 and Eisenhoffer et al., 2017), while the basal p63-positive cells generated apoptotic bodies (Reviewer Figure 1), consistent with our current studies. Importantly, these data revealed that basal p63-positive cells exhibit similar apoptotic morphologies to death induced by the NTR/MTZ system. Yet, this assay was limiting in that the cells did not die in a stereotyped location or at a specific time after UV exposure, and therefore, made filming engulfment of the resulting apoptotic material by neighboring stem cells difficult. Further, the dying stem cells did not have a differential fluorescent label that would permit tracking of the resulting ABs, as is currently the case with our

Reviewer Figure 1

Reviewer Figure 1. UV-induced damage of the zebrafish epidermis. Maximum intensity projection confocal image of a zebrafish larvae immunostained for activated caspase-3 (scale = 50 μ m). This analysis revealed that UV-induced damage promotes both the extrusion of surface epithelial cells (arrowheads) and generation of apoptotic bodies by the basal stem cells. (asterisks).

NTR/MTZ based system. We are continuing to refine this method and plan to use this for future assays.

Minor points:

4. Figure legends in the main text are often not detailed enough to explain figures. For example, Figure 5 j: what do SSC and FSC stand for? In Figure 5 i: what is the control? Figure 3 k': what is the asterisk indicating?

We have clarified this point in the legend for Figure 5 by defining SSC (side scatter), FSC (forward scatter) and stating that the control (grey area in the FACS plot in Figure 5i) are 1-5 μ m extracellular vesicles isolated from zebrafish larvae under homeostatic conditions, and therefore, not from epithelial stem cell-induced apoptosis. Likewise, we have also amended the legend for Figure 3 to denote the label colors and that the asterisk is labeling the engulfing cell.

5. Some figures are missing information that would aid understanding. Figure 3 a-h: label colours; Figure 4 a-f: label colours.

We have added labels that represent the respective colours in Figures 3 a-h and 4 a-f.

6. Figure 3 j-l: how are individual GFP+ cells counted? From the images provided, it appears that the GFP is too weak to easily identify cell borders, and nuclei are not labeled.

To quantify the Tcf-positive cells, we analyzed both live imaging datasets and fixed specimens. While the GFP is indeed weak, the sparse nature of the signal makes identifying individual cells straightforward. We confirmed this approach in the fixed tissue analysis by co-staining with DAPI to label nuclei and ensure that individual cells were quantified.

7. Statistical methods: the use of the T-test is inappropriate when comparing more than two groups (i.e. for quantifying BrdU- or caspase-positive cells). Use ANOVA when comparing multiple groups.

T-tests were used to compare vehicle treated or untreated controls versus the MTZ treated larvae. We have also performed ANOVA analyses to compare multiple groups. We thank the reviewer for bringing this point to our attention and have clarified this point where relevant in the figure legends.

8. For the Wnt8 Crispr, the rationale behind using only the injected fish is not explained. Is it because the stable line gives a gastrulation phenotype like described Wnt8 morphants? If so, the rationale should be better explained. Furthermore, the materials and methods indicates that exons 2 and 5 were targeted using a single guide RNA for each. However, it is not clear in the main text or in the figure legends which exon was targeted in the examples shown (although Supplementary Figure 5 shows indel frequencies for exon 2 only), nor is it stated whether similar results were obtained for both exons.

Stable mutant lines for Wnt8a did not exist when we started these studies. Our studies also required alteration of Wnt8a in a large number of larvae for the purification of ESABs and examination of Wnt8a. Our analysis showed that microinjection of guide RNAs (+ Cas9 protein) for both exon 2 and 5 animals, or each guide independently, achieved similar results and recapitulated the gastrulation and altered posterior axis development phenotypes previously reported using antisense morpholino oligonucleotides. Targeted barcoded sequencing, as well as traditional Sanger sequencing and Tracking of Indels by Decomposition (TIDE) analysis, yielded similar indel frequencies for both exon 2 and 5. This point has been updated in the methods section. We have since worked to generate stable novel mutant alleles of Wnt8a at both exon2 and exon 5. Thus far, compound transheterozygous crosses of F1 animals with a mutation at exon 2 show a similar reduction in apoptosis-induced proliferation. The unique mutant alleles in the F2 generation are currently growing in our facility, but are not yet old enough to breed to perform further analysis at this time.

9. The rationale for the choice of the Wnt genes examined by RT-PCR should be better explained.

We initially choose the candidate Wnt genes examined by RT-PCR based on those that have been previously implicated in stem cell activity (2b, 4a, 3a) and to distinguish between canonical (2b, 3a, and 8a) and non-canonical pathways (5b and 11). To help clarify this point to the reader, we have added a sentence to page 7 stating that “We examined components of both the canonical and non-canonical Wnt signaling pathway, as well as those previous shown to be involved in epithelial stem cells or influenced by damage”.

10. Were the other upregulated Wnts (Wnt4b, Wnt11) localized to apoptotic cells or not? It is not clear in the text why Wnt4b and Wnt11 were excluded from further analysis.

We have been unable to confirm if Wnt4b and Wnt11 are localized to apoptotic cells due to a lack of available reagents. We are currently having zebrafish specific antibodies against Wnt4b and Wnt11 generated, as well as pursuing in situ hybridization to detect changes in expression of Wnt4b and Wnt11 and determine if these transcripts are localized to the apoptotic cells.

11. Figure 3 d-e: Is the dying cell really a basal cell? It looks like it could be a peridermal cell.

Under homeostatic conditions, we found that periderm cells can be easily distinguished from basal cells by A) their apical position in the tissue B) the presence of microridges and C) the difference in electron dense contrast within the basal cells. Our previous work has shown that when apically located periderm cells die, they are eliminated apically from the tissue into the media. For dying cells, we used similar criteria. The images in Figure 3 d-e clearly show a basally located cell (sitting atop the basement membrane) with intact periderm cells (with a lighter color and having microridges) above. These were not observed in wild-type or vehicle control (DMSO) treated embryos, or in animals in which we induced extrusion of the periderm cells.

12. The authors have previously shown that inducing apoptosis in zebrafish embryos by treatment with G418 leads to apical extrusion of peridermal cells, rather than engulfment by

neighboring cells. Why is the outcome so different here? This is alluded to briefly in the results, but should be explored further in the discussion.

This is actually how this project started out: We asked if apoptotic stem cells extrude from the epithelium, much like we had previously found for the periderm cells. Previous studies have also found that dying melanocytes, located below the periderm, are extruded from the body of the zebrafish into the surrounding media (Parichy et al, 1999). To our surprise, we found the apoptotic stem cells rounded up and did not extrude from the tissue. Instead, the dying cells generated apoptotic bodies that were engulfed by neighboring epithelial stem cells. We have been able to specifically induce elimination of periderm cells by extrusion using NTR/MTZ system, but we have not yet examined if these cells produce Wnt signals after damage during the elimination process. Therefore, while the cell type or location specific response to damage is a fascinating question, the lack of available data makes us uncomfortable discussing comparisons or differences between the two paradigms at the current time.

13. In the Discussion, the authors might want to speculate on the nature of the (cell death-dependent) cofactor, which is required by Wnt8a to exhibit its pro-proliferative gain-of-function effect.

One hypothesis for a cell-death dependent cofactor is the process of phosphatidylserine externalization during apoptosis. It may be possible that Wnt8a localized at the cell cortex may also be externalized to the surface of apoptotic cells during apoptosis. Such a mechanism would re-localize new or existing Wnt8a ligands to the plasma membrane, allowing them to activate Wnt-mediated apoptosis-induced proliferation. The PS externalization data in Supplementary Figure 6a-f is intriguing and supports this idea, but we felt more experimental evidence was required before speculating on this putative mechanism.

14. Please cite the work by Fischer et al. (PLoS Genet 2014), pointing to the existence of caspase-dependent mitogens generated by “naturally dying” outer keratinocytes of zebrafish breeding tubercles to stimulate compensatory proliferation of basal keratinocytes, assuring proper homeostasis of these self-renewing epidermal appendages.

We have now cited Fischer et al, 2014 as an example of caspase-dependent mitogens generated by “naturally dying” outer keratinocytes of zebrafish breeding tubercles to stimulate compensatory proliferation of basal keratinocytes.

Summary of changes to the manuscript

All changes to the main text and supplementary material have been highlighted and are described below.

Changes to the main text

1. A new sentence has been added to the introduction on page 4 stating that dying outer keratinocytes in the zebrafish breeding tubercles stimulate compensatory proliferation of basal p63-positive cells to ensure homeostasis, and refer to Fischer et al. (PLoS Genet 2014).
2. A sentence was added on page 6, stating that the large majority (97.87% of 1506 BrdU+ cells analyzed) of dividing cells after induced apoptosis were p63 positive.
3. Also on page 6, a sentence was added to clarify that caspase activity in regulating non-apoptotic processes was not affected by treatment with the apoptosis inhibitor.
4. Figure 1K was modified to incorporate new data using the caspase 3 inhibitor zDEVD-fmk and further support the role of cell death in promoting proliferation.
5. On page 7, a sentence has been added stating that we examined components of both the canonical and non-canonical Wnt signaling pathway, as well as those previous shown to be involved in epithelial stem cells or influenced by damage.
6. Figure 2C was modified to incorporate new data using the caspase 3 inhibitor zDEVD-fmk to support the idea that Wnt8a is produced in the apoptotic cells in a caspase-dependent manner.
7. On page 8, we changed the sentence to state that “cellular extensions containing the F-actin marker Lifeact”, and removed the portion about the process being actin-mediated.
8. On page 8-9, we have added a sentence describing new data (Supplemental Fig. 4d-f) investigating the role of the macrophages in the clearance of the apoptotic bodies.
9. On page 9, we modified the sentence describing pharmacological inhibition of Wnt signaling to state: “Pharmacological inhibition of Wnt signaling, using a tankyrase inhibitor IWR-1”.
10. Figure 4 was modified to incorporate color-coded labels for both BrdU incorporation and the p63-positive basal cells.
11. A sentence was also added to page 11 that describes our new evidence (Supplementary Figure 6) that phosphatidylserine (PS) is externalized to the outer leaflet of the plasma membrane and that purified ESABs have a large number of PS positive foci on the surface.
12. The codes for the respective colors in the images in Figures 3 and 4 have been added to the legend. Also, the legend for Figure 5 has been adjusted to state that the grey control areas for

Fig. 5i refer to extracellular vesicles isolated from zebrafish larvae under homeostatic conditions, and a correction has been added to state that the scale bar in Fig. 5k is 500nm.

Changes to the supplementary material

1. Figure S1d was changed to include the quantification of the number of caspase 3 positive cells after treatment with the caspase-3 inhibitor zDEVD-fmk.
2. Arrowheads were added to Figure S1e to denote the bulges in the surface epithelium.
3. Images and quantification of caspase-3 activity were moved to S3d-e to accommodate the addition of the macrophage data.
4. Visualization, tracking and quantification of macrophage activity after induced epithelial stem cell apoptosis was added to S4d-f.
5. Whole mount immunohistochemical data, and immunogold labeling on purified ESABs, demonstrating that phosphatidylserine is externalized and present on the surface of apoptotic cells and bodies was added to S6a-f.

REVIEWERS' COMMENTS:

Reviewer #1 (Remarks to the Author):

The authors have adequately addressed all my questions and concerns.

Reviewer #3 (Remarks to the Author):

I already liked the first version of the manuscript very much, and the revised version even more. In the revised manuscript and/or the rebuttal letter, the authors have very satisfactorily addressed all of the points raised in my former review. They have successfully carried out most of the proposed additional experiments, while convincingly explaining why some of the other proposed additional data or discussions are beyond the scope of this work, too preliminary and/or too speculative to be included in the manuscript. Clearly, with the new data that have been added to the manuscript, the manuscript has significantly improved by providing further evidences for the former conclusions. Therefore, I strongly recommend the acceptance of the manuscript for publication in Nature Communications without further revisions.

REVIEWERS' COMMENTS:

Reviewer #1 (Remarks to the Author):

The authors have adequately addressed all my questions and concerns.

Reviewer #3 (Remarks to the Author):

I already liked the first version of the manuscript very much, and the revised version even more. In the revised manuscript and/or the rebuttal letter, the authors have very satisfactorily addressed all of the points raised in my former review. They have successfully carried out most of the proposed additional experiments, while convincingly explaining why some of the other proposed additional data or discussions are beyond the scope of this work, too preliminary and/or too speculative to be included in the manuscript. Clearly, with the new data that have been added to the manuscript, the manuscript has significantly improved by providing further evidences for the former conclusions. Therefore, I strongly recommend the acceptance of the manuscript for publication in Nature Communications without further revisions.